

# CapsNets continuing the convolutional quest

**Sascha Diefenbacher[1], Hermann Frost[2], Gregor Kasieczka[2],
Tilman Plehn[1]\* and Jennifer M. Thompson[1]**

**1** Institut für Theoretische Physik, Universität Heidelberg, Germany
**2** Institut für Experimentalphysik, Universität Hamburg, Germany

\* plehn@uni-heidelberg.de

## Abstract

Capsule networks are ideal tools to combine event-level and subjet information at the LHC. After benchmarking our capsule network against standard convolutional networks, we show how multi-class capsules extract a resonance decaying to top quarks from both, QCD di-jet and the top continuum backgrounds. We then show how its results can be easily interpreted. Finally, we use associated top-Higgs production to demonstrate that capsule networks can work on overlaying images to go beyond calorimeter information.



## Content

## 1 Introduction

New developments in machine learning have recently started to transform different aspects of LHC physics. The most visible development is, arguably, deep learning in subjet physics. The

underlying idea is to replace multi-variate analyses of high-level observables by deep neural networks with low-level observables. It follows directly from our improved understanding of subjet physics both experimentally and theoretically, and from the rapid development of standard machine learning tools [1].

A standard approach to deep learning of jet physics is based on jet images, where we extract information from heat maps in the rapidity vs azimuthal angle plane [2–5]. Several studies have looked at what information a neural network can extract from jets [6–10]. The most relevant measurements come from the calorimeter and need to be combined with tracking information. Standard benchmarks based on jet images or alternative network setups are quark-gluon discrimination [11–16], $W$-tagging [17,18], Higgs tagging [19,20], and top-tagging [21–28]. This relatively straightforward classification task only served as a first attempt of deep learning in LHC analyses [29], and the progress in this field should encourage us to search for more challenging and transformative applications. One promising line of reserach is related to ways of training neural networks at the LHC, for instance using weakly supervised learning [30–34], unsupervised classification [35], or unsupervised autoencoders [36–39]. Alternatively, we can extend our classification task trained on data to include statistical and theoretical uncertainties [40].

Once we have sufficient control over the network training we can come back to modern LHC physics, where jets have turned from the main objects of event analyses to a somewhat arbitrary separation line between subjet analysis and event-level analyses. The question becomes how neural network architectures cope with the full event information. We emphasize that such event information should again be low-level observables rather than a small number of 4-vectors describing the hard process at this high level [41]. A natural extension of convolutional networks on event and jet images [42,43] are capsule networks [44,45]. For applications in astrophysics, see *e.g.* Ref. [46,47]. The main advantage of capsules is in analyzing structures of objects and simultaneously their geometric layout. It perfectly matches our task of combining subjet information with the event-level kinematics of jets and other particles.

In this paper we start with a brief introduction to capsule networks as an extension of convolutional networks in Sec. 2. Next, we apply capsule networks to the classification of di-top events at the subjet level in Sec. 3. This allows us to benchmark our capsule network with established machine learning top taggers using for example convolutional networks [26,27]. Next, we separate full events corresponding to a $Z'(\to t\bar{t})$ signal from $t\bar{t}$ and from di-jet backgrounds in Sec. 4. Here we introduce multi-class capsules to control the different backgrounds. In Sec. 5 we how these results can be visualized especially well. Finally, in Sec. 6 we consider a challenging application, the semi-leptonic final state of $t\bar{t}H_{bb}$ production. It allows us to explore the full power of capsule networks to extract information from overlaying images, going beyond calorimeter images and opening a path towards tracking information [48].

We emphasize that in this study we do not discuss the issues related to training networks for event tagging or the systematic and theoretical uncertainties related to it. They need to be tackled now that we know how large event images can be efficiently analyzed by capsule networks — as the natural extension of convolutional networks working on jet images.

## 2 Capsule networks

In this paper we introduce capsule neural networks (CapsNets) as a natural replacement for standard convolutional neural networks in LHC physics. We refer to the established convolutional networks as scalar CNNs because they rely on single numbers. CapsNets replaces these single numbers with capsule vectors describing the feature maps. For example, 24 feature

maps with $40 \times 40$ entries each could be defined as 1600 capsules of dimension 24, or 3200 capsules of dimension 12, or 4800 capsules of dimension 8, etc. Each capsule can be thought of as a vector in signal or background feature space, depending on which it describes. The length of this vector then encodes how signal-like or background-like the image is. The idea behind these vectors is that they can track the actual geometric position and orientation of objects, which is useful for images containing multiple different objects. In particle physics, an entire event image is a perfect example of this.

Just like a scalar CNN, a CapsNet starts with a pixelled image, in our case the calorimeter image of a complete event with $180 \times 180$ pixels. This image is analyzed with a convolutional filter, for example extracting edges. The size of these kernels is not fixed, so one way of reducing the size of a sparsely filled image is to choose kernels with at least $(n + 1) \times (n + 1)$ pixels and to move $n$ rows or columns per step. This is known as a convolution with stride $n$, in contrast to pooling layers which simply decrease the resolution of the image. How significant the difference is between these approaches depends on the details of the analysis [44, 45, 49]. Our CapsNets include several layers of convolution with multiple feature maps. They extract the relevant information from the input image, and is so far identical to a scalar CNN. The advantages and power of the CapsNet come from the additional capsule layers after the convolutions.

Deep CapsNets consist of several capsule layers. After the convolution part of the CapsNet, each layer consists of a number of parallel capsules. These capsules have to transfer information matching their vector property. In Fig. 1 we illustrate a small, two-layer CapsNet with three initial capsules $\vec{x}^{(j)}$ of dimension two linked through routing by agreement [44] to four capsules, also of dimension two,

$$x_i^{(j)} \longrightarrow v_{i'}^{(j')} \quad \text{with} \quad i = 1, 2 \quad i' = 1, 2 \quad j = 1, 2, 3 \quad j' = 1, 2, 3, 4 . \tag{1}$$

For deeper networks the dimensionality of the resulting capsule vector can, and should, be larger than the incoming capsule vector. However, in our illustration we keep the dimensionality of the capsules at two for clarity. To get from three to four capsules we first define four combinations of the three initial capsules. Their entries are defined as $u_{i'}^{(j,j')}$, and they are related to the initial capsule vectors $\vec{x}^{(j)}$ through trainable weights,

$$u_{i'}^{(j,j')} = \sum_{i=1,2} w_{i'i}^{(j,j')} x_i^{(j)} , \tag{2}$$

as indicated by the arrows in Fig. 1. The assignment of lower and upper indices in our description only serves illustrational purposes. Next, we need to contract the index $j$ to define the four outgoing capsules. For this purpose we define another set of trainable weights and write

$$v_{i'}^{(j')} = \sum_{j=1,2,3} c^{(j,j')} u_{i'}^{(j,j')} . \tag{3}$$

These weights $c^{(j,k)}$ get normalized through a SoftMax operation

$$\sum_{j'=1,2,3,4} c^{(j,j')} = 1 \quad \forall j$$

$$c^{(j,j')} = \text{SoftMax}_{j'} \, c'^{(j,j')} = \frac{\exp c'^{(j,j')}}{\sum_\ell \exp c'^{(j,\ell)}} , \tag{4}$$

on a set of general weights $c'$. This ensures that the contributions from one capsule in the former to each capsule in the current layer add up to one. Furthermore, the squashing step

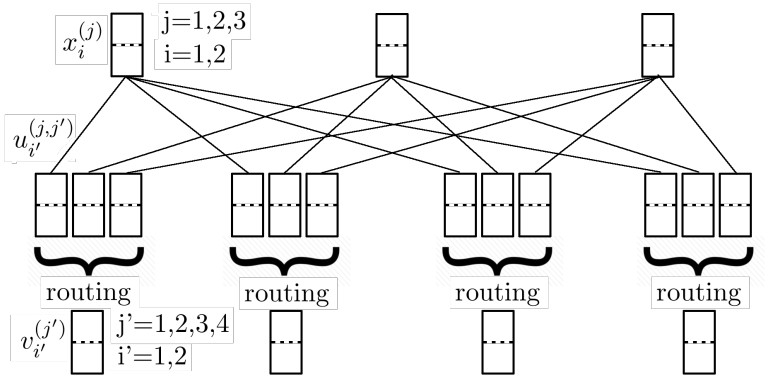

Figure 1: Sketch of a CapsNet module with two simple capsule layers.

applied after each capsule layer ensures that the length of each output capsule vector remains between 0 and 1,

$$v_{i'}^{(j')} = \vec{v} \to \vec{v}' = \frac{\vec{v}^2}{1 + \vec{v}^2} \, \hat{v} \tag{5}$$

$$\vec{v} \to \vec{v}' = \frac{|\vec{v}|}{\sqrt{1 + |\vec{v}|^2}} \, \hat{v} \, , \tag{6}$$

with $\hat{v}$ defined as the unit vector in $\vec{v}$-direction. The advantage of Eq.(6) over Eq.(5) is that it does not iteratively shrink small inputs to zero. This feature becomes important when considering multiple capsule layers.

Up to now we have constructed a set of four capsules from a set of three capsules through a number of trainable weights, but not enforced any kind of connection between the two sets of capsule vectors. We can extend the condition of Eq.(4) to consecutively align the vectors $\vec{u}^{(j,j')}$ and $\vec{v}^{(j')}$ through a re-definition of the weights $c^{(j,j')}$. This means we compute the scalar product between the vector $\vec{u}^{(j,j')}$ and the squashed vector $\vec{v}^{(j')}$ and replace

$$c'^{(j,j')} \longrightarrow c'^{(j,j')} + \vec{u}^{(j,j')} \cdot \vec{v}^{(j')} \, , \tag{7}$$

which converges once $\vec{u}^{(j,j')}$ and $\vec{v}^{(j')}$ are parallel. We apply this replacement to each capsule, or fixed $j'$, individually, before we once again apply the SoftMax operation. We repeated this for 3 routings, which has been shown in other studies to give the best results [44]. The routing is illustrated in Fig. 2, where the blue vectors represent the three intermediate $\vec{u}^{(j,j')}$ in each set and the red vector is the combination $\vec{v}^{(j')}$. We can see how, with each routing iteration, the vectors parallel to $\vec{v}^{(j')}$ become longer while the others get shorter.

In the CapsNet framework we use the squashed length of the output vectors $\vec{v}^{(j')}$ for classification. In complete analogy to the scalar CNN we differentiate between signal and background images using two output capsules. The more likely the image is to be signal or background, the longer the output capsule vectors will be. The corresponding margin loss for a set of output capsules $j'$ is defined as

$$L = \sum_{j'} L^{(j')}$$

$$L^{(j')} = T^{(j')} \max\left(0, m_+ - |\vec{v}^{(j')}|\right)^2 + \lambda(1 - T^{(j')}) \max\left(0, |\vec{v}^{(j')}| - m_-\right)^2 \, . \tag{8}$$

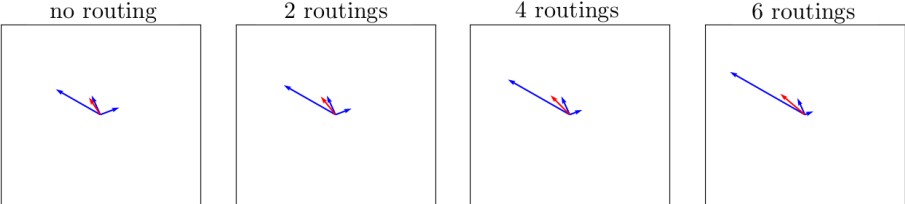

Figure 2: Effects of the routing/squashing combination. In blue we show the intermediate vectors, in red we show the output vector after squashing.

$T^{(j')}$ is the truth label of the input, so for a simple classification task we use $T^{(1)} = 1$ and $T^{(2)} = 0$ and the loss function consists of the two terms

$$L^{(1)} = \max\left(0, m_+ - |\vec{v}^{(1)}|\right)^2$$
$$L^{(2)} = \lambda \max\left(0, |\vec{v}^{(2)}| - m_-\right)^2 . \tag{9}$$

Using, for example, $m_+ = 0.9$ means that the network will seek signal vectors $\vec{v}^{(1)}$ with length above 0.9, where the loss vanishes. Similarly, for $m_- = 0.1$ the network prefers background vectors $\vec{v}^{(2)}$ shorter than 0.1. While these target numbers of the capsule length, 0.9 and 0.1, sum up to unity, nothing forces the actual length of all capsules in a prediction to do the same. Using the $\lambda$ parameter we can scale the importance of the two terms in the loss function. We chose $\lambda = 0.5$, putting the main emphasis on having the correct capsule length being close to the target number. Although not relevant for the conclusions in this study, the CapsNet receives an additional term in its loss function from the reconstruction of the initial image from the outputs.

For a 2-class classification task it should be possible to define a simpler setup where one output capsule encodes the entire signal vs background information. This also means that for our setup the capsule length output $|\vec{v}^{(i)}|$ cannot be linked to a probability, but as a set of scores which describe the how signal-like or background-like an event is. Combining them into a single classifier is not unique, as we will see later. We will also see that the LHC analyses we propose in this paper are not 2-class classification tasks, so we keep our multi-capsule output for now.

The advantage of the CapsNet over the scalar CNN is that each entry of the capsule vector can learn certain features independently of the other entries, and only the combination of all entries is required to separate signal and background. This flexibility should, for instance, replace the pre-processing which is often used for jet images [26,27]. Following the original test cases for capsule networks, the vector entries can also learn individual patterns independently from the geometric arrangement. This is precisely what we will exploit in our combination of subjet and event-level patterns in LHC events. We implement all our CapsNets with the KERAS PYTHON package [50] and a TENSORFLOW [51] back-end. We also make use of the usual ADAM optimizer [52].

# 3  Di-top tagging

Before we use CapsNets to analyze full events, we need to confirm that they successfully analyze subjet structures. As an experimentally and theoretically safe, established benchmark we use top tagging [29, 53–57], specifically the signal process

$$pp \rightarrow Z' \rightarrow t\bar{t} \tag{10}$$

for $m_{Z'} = 1$ TeV and a narrow width of $\Gamma_{Z'} = 1$ GeV. A small width is useful when we eventually extract the narrow resonance from a continuum background. Because this is a BSM process, we first generated the model with FEYNRULES [58, 59]. This simplified $Z'$ model extends the Standard Model Lagrangian by

$$\mathcal{L}_{Z'} = c_1 \sum_q c_q \bar{q} \gamma_\mu q Z'^\mu \,, \tag{11}$$

where $c_1$ and $c_q$ are freely chosen constants determining the normalization of the signal. The $Z'$ decays to a $t\bar{t}$ pair, which in turn decay hadronically.

Such a $t\bar{t}$ resonance search allows us to split the analysis into two steps [60]. First, we focus on the subjet-level information from the two fat jets and ignore their event-level kinematics. For this purpose we limit ourselves to the light-flavor di-jet background

$$pp \to jj \qquad \text{with } j = g, u, d, c, s, b \,, \tag{12}$$

and will add the continuum top pair background in the next section.

All events in this study are for a 13 TeV LHC. We generate leading-order events with SHERPA2.2.5 [61] and the COMIX matrix element generator [62], where we enable parton shower and hadronization effects. We neglect underlying event and pile-up, which we assume can be removed by dedicated tools [63–65]. We use the 5-flavor LO NNPDF3.0 PDF set [66]. Detector effects are simulated with DELPHES [67] and a standard ATLAS card with a modified jet radius and jet algorithm for each process. All jets are defined by FASTJET [68, 69]. In this section jets are defined by the C/A jet clustering algorithm [70], with $R = 1.0$ and

$$p_{T,j} > 350 \,\text{GeV} \qquad \text{and} \qquad |\eta_j| < 2.0 \,. \tag{13}$$

From this output we extract the calorimeter hits and transform them into a 2D jet image with $E_T$ as the pixel value. The calorimeter images have a size of

$$180 \times 180 \text{ pixels,} \tag{14}$$

covering $|\eta| < 2.5$ and $\phi = 0 \dots 2\pi$. The periodicity in $\phi$ is accounted for by phi-padding with an appropriate depth or number of repeated pixels in both positive and negative $\phi$ direction. We separately choose the amount of padding for each convolutional layer and equal to half the respective kernel size. For this benchmarking exercise we then remove all event-level information, such as $\eta$ and $\phi$ positions of the jets. We take the event-level calorimeter images, pad them with zeros in $\eta$ and symmetrically in $\phi$ and select $40 \times 40$ pixel sections around the axes of the two leading jets. Two such jet images are then pasted back into empty $180 \times 180$ images. This process is illustrated in Fig. 3.

These simplified $(180 \times 180)$-pixel event images are what we feed into our CapsNet shown in the top panel of Fig. 4. Our architecture avoids pooling and instead uses convolutions with stride two, as outlined in Sec. 2. We produce 32 feature maps and for the first two layers we use a $9 \times 9$ kernel and a stride of two. Then we reduce the kernel size to $5 \times 5$ for the third layer, still with a stride of two. Finally, we apply one regular convolution with a stride of one and a $3 \times 3$ kernel. With this final convolution we also increase the number of feature maps to 96.

Analogous to the original capsule paper [45], we transition between the convolutional and capsule parts by re-shaping the output of the convolutional layers into a capsule layer with $j \leq 5888$ capsules of dimension $i \leq 6$ and add a second layer with dimension $i' \leq 8$ and $j' = 1, 2$ capsules, which are used as outputs for the classification. Here $i^{(')}$ and $j^{(')}$ run over the dimensionality and number of capsules, respectively, as described in Sec. 2.

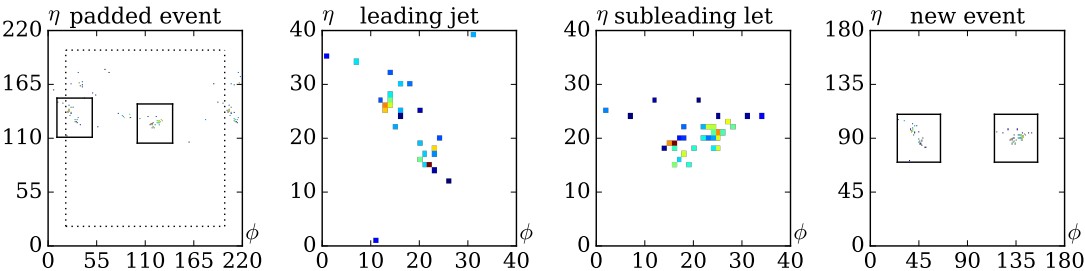

Figure 3: Processing of the event images to a pair of top images.

If we want to evaluate the performance of our network we need an estimator to build an ROC curve. In a scalar CNN with SoftMax activation in the final layer we usually use the output of the signal neuron, because the background neuron does not give an independent result. The equivalent approach for our CapsNet could rely on the length of the signal capsule, $|\vec{v}^{(S)}|$. However, our CapsNet does encode additional information in the background capsule, so based on the output capsules in Eq.(8) we can define estimators of the kind

$$|\vec{v}^{(S)}| \qquad \text{or} \qquad |\vec{v}^{(S)}| - |\vec{v}^{(B,1)}| - |\vec{v}^{(B,2)}| - \cdots \tag{15}$$

This choice affect the tagging performance for realistic training. Throughout the paper we will use the second choice as the default.

We can now compare the performance of our CapsNet to a combination of two scalar CNN taggers, specifically the Rutgers DEEPTOP tagger [26, 27]. In contrast to the minimal pre-processing we use for the event image capsule network, for the Rutgers tagger and the jet image capsule network we employ the full pre-processing for each jet as described in Ref. [27]. The jets are selected and centered around the $p_T$ weighted centroid of the jet, and rotated such that the major principal axis is vertical. The image is then flipped to ensure that the maximum activity is in the upper-right-hand quadrant. Finally, the images are pixelated and normalized.

It is shown in the center panel of Fig. 4. We use a total of 500,000 events, split into three parts training and one part each for testing an validation. For training we use the ADAM optimizer with a learning rate of 0.001 and a decay of 0.9, and we employ early stopping to interrupt training once the validation accuracy stops increasing. The result of this comparison can be seen in the left panel of Fig. 5. The shaded curve represents the two estimators given in Eq.(15), where in this case the signal capsule alone gives the better results and an ROC curve more compatible with the 2-class scalar CNN. Within this uncertainty, the CapsNet performs as well as two copies of a dedicated tagger for the subjet information alone. Given that the CNN is well-optimized, this is the best we can expect for our relatively straightforward CapsNet.

To allow for a direct comparison with many other tools, we also apply our CapsNet to single top jets from a public dataset [29]*, based on events generated for the study in Ref. [28]. In that case there exists no event-level information for the CapsNet shown in the bottom panel of Fig. 4. Again, the CapsNet turns out competitive with state-of-the-art convolutional networks, albeit not quite with the leading tools presented in Ref. [29].

# 4    Di-top resonance

In our second benchmarking step we now consider event-level information. To see how the CapsNet uses event kinematics in addition to subjet-level information we first study the con-

---

*link to top tagging sample, for more information and citation please use Ref. [28].

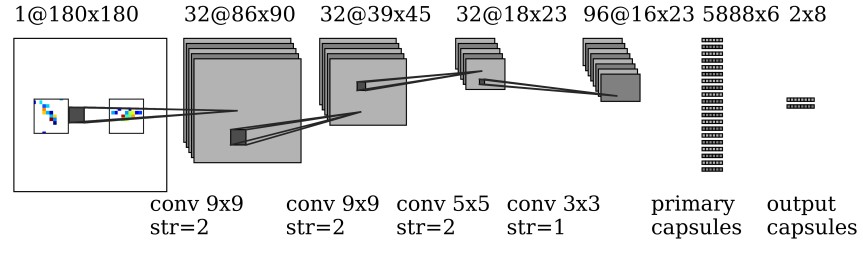

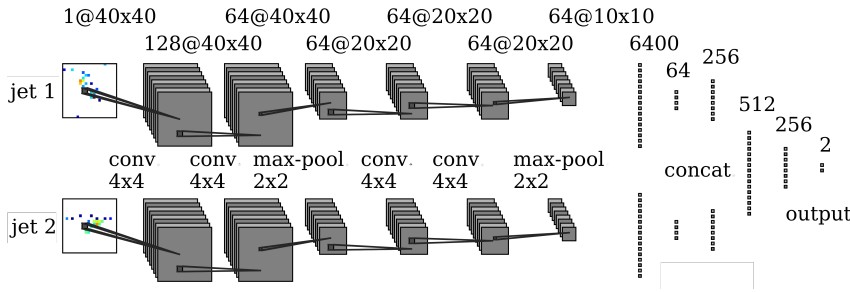

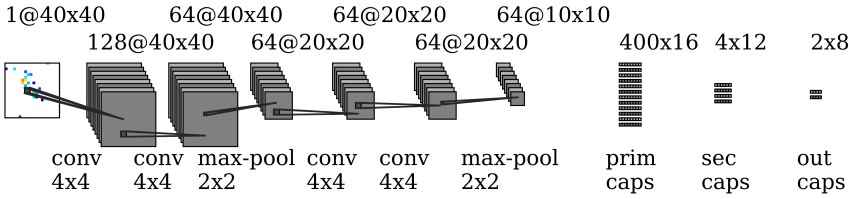

Figure 4: Top: convolutional CapsNet architecture based on stride-convolutions, used for di-Top tagging. Center: scalar CNN di-top tagger using the Rutgers DEEPTOP architecture [26, 27]. Bottom: CapsNet single top jet tagger architecture.

tinuum $t\bar{t}$ background to our $Z'(\to t\bar{t})$ signal,

$$pp \to t\bar{t} \qquad \text{(SM)}. \tag{16}$$

For this classification the subjet information does not help. Only the combination of the continuum $t\bar{t}$ and QCD backgrounds then requires the CapsNet to learn both the geometry of the event and the subjet differences of top and QCD jets.

Now that the signal jets and background jets are both boosted top quarks, the network needs to rely on the $Z'$ kinematics and differences in radiation patterns between signal and background. On the CapsNet side the increased complexity of the full events leads to a slightly more involved architecture than the one described in Sec. 3. Our new architecture combines max-poolings, average-poolings, and convolutions, to make it easier to (also) focus on large-scale features. It is shown in Fig. 6. The idea behind the setup is that (i) the max-pooling preserves the highest value pixels, allowing the network to learn both the absolute and relative jet positions, and (ii) the average-pooling preserves the total transverse energy. Consequently, we use two different resolutions, $45 \times 45$ to learn the jet $E_T$ and $9 \times 9$ to learn the total energy in the event. The two different pooling strategies are implemented as two parallel branches in the network. The average pooling branch is further subdivided into three branches. This allows for three different kernel sizes to be used in parallel, corresponding to three different scales of activity.

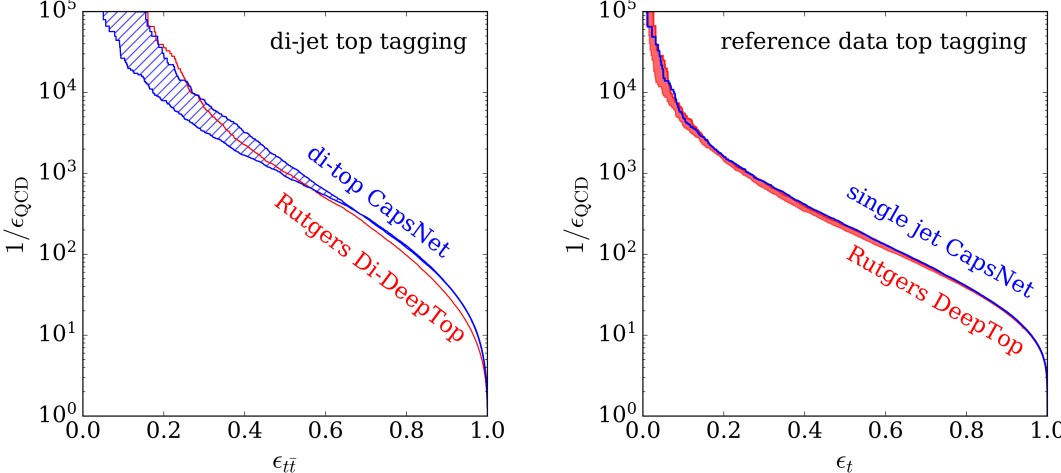

Figure 5: Left: comparison between CapsNet and the Rutgers DEEPTOP CNN for for the di-tops vs QCD di-jets. Right: same comparison for single top jets using a standard dataset [29]*.

As the benchmark for the event-level analysis we use a boosted decision tree with the SCIKIT-LEARN PYTHON package [72] and using ADABOOST [73]. No cuts are placed on the $t\bar{t}$ final state, and the events are generated according to the procedure described in Sec. 3. We give the BDT the event-level information

$$\{m_{jj}, p_{Tj_1}, p_{Tj_2}, \eta_{j_1}, \eta_{j_2}\}. \tag{17}$$

The BDT has a maximal depth of three and uses 100 estimators. Training and testing of the BDT are performed with the same samples used for the network training and evaluation. We use 500,000 events, split into 300,000 training events, 100,000 testing events and 100,000 validation events.

Figure 7 shows that our extended CapsNet architecture performs significantly better than both the simpler CapsNet and the BDT baseline. Specifically, the convolutional CapsNet slightly under-performs the BDT, whereas the larger architecture is more able to describe the complexity of a full event, leading to a significant improvement over a simple BDT approach.

Combining these results with those of the previous section, we now have all the building blocks to discriminate $Z' \to t\bar{t}$ signal events from the mixed QCD+$t\bar{t}$ background. We can follow two different approaches: consider the problem as signal vs background classification or think of it as classifying events into one signal and two background categories. For this comparison, we use the pooling setup shown in Fig. 6, as well as the convolution setup from Fig. 4. Moving from one common background label to two distinct backgrounds leads us to a multi-class CapsNet, including the choice of estimators alluded to in Eq.(15). In training the 2-class network we use a sample with half signal and half background events, the background further divided evenly between $t\bar{t}$ and QCD background . For the 3-class case we use equal parts for each label. We considered splitting the backgrounds following their respective rates, but in this case the background sample would have been entirely dominated by QCD. As in the previous sections the training sample consists of 300k events combined.

The the two panels of Fig. 8 show the rejection of the QCD background (left) and the continuum $t\bar{t}$ background (right) by CapsNets trained on mixed background samples. The number of classes used has different effects for the two architectures.

For the convolutional architecture there appears to be no significant difference between the two-class and three-class versions. This is because the convolutional setup is very apt at

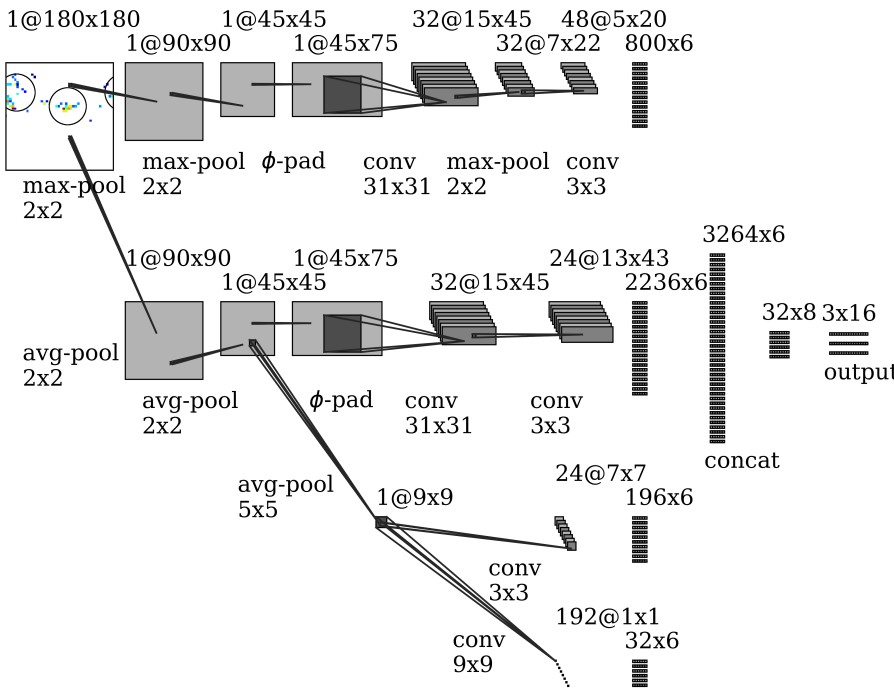

Figure 6: Pooling CapsNet architecture. The re-shaping of the final convolution results into capsules with dimension $i \leq 6$ is not detailed.

extraction subjet features, but not good at event-level information. Given that the combined background capsule of the two-class setup encodes subjet features efficiently, a dedicated QCD capsule offers little improvement. Consequently, the convolutional CapsNet performs very weakly for the $t\bar{t}$ background rejection, and moving from two to three classes helps very little with this structural deficit.

The situation is different for the more carefully constructed pooling CapsNet. In QCD background rejection it very clearly benefits from the 3-class setup. The reason is that the pooling setup is designed with event-level kinematics in mind, so when one capsule faces both backgrounds it will focus on the event-level features and deliver a poor QCD background rejection. In its 3-class version the pooling setup can train a dedicated QCD capsule on the subjet features extremely well. For the $t\bar{t}$ background rejection the pooling CapsNet the third class leads to no improvement, because the 2-class network already learns the event-level information.

Altogether, we find that a 3-class pooling CapsNet is best suited for extracting the $t\bar{t}$ resonance signal from a mixed $t\bar{t}$ and QCD background. When comparing its performance to the that from a pure $t\bar{t}$ background in the right panel of Fig. 8, we still notice a slight drop in performance for the $t\bar{t}$ background rejection. This has two contributing factors: first, the network needs to learn 50% more features in going from a 2-class to a 3-class problem with the same number of weights. Second, the additional output class adds more possibilities for mis-stating. The first issue can be fixed by adding more weights up to the point where computing power becomes the limiting factor, the second is inherent to multi-class problems.

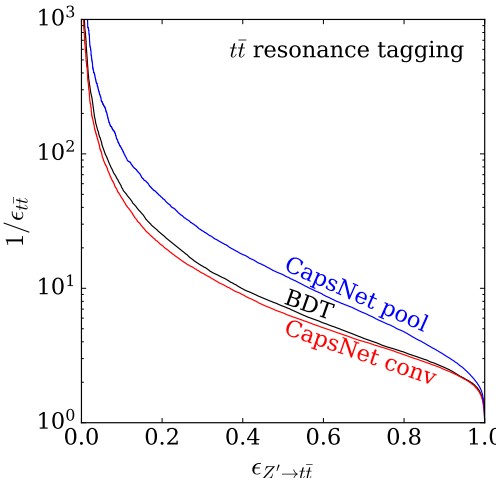

Figure 7: ROC curves for two capsule networks and the BDT benchmark, trained and tested on $Z'(\to t\bar{t})$ signal and continuum $t\bar{t}$ background events.

## 5    Inside capsules

Before moving to even more complex problems, we want to understand what the capsule vectors learn. For this visualization aspect we again separate $Z'(\to t\bar{t})$ from QCD di-jet events. The signal and background events then differ in event-level kinematics and in jet substructure. To further simplify the problem we reduce the resolution of input images from $180 \times 180$ to $45 \times 45$ pixels using sum-pooling with a kernel size 4. This brings us close the size of MNIST digits of $28 \times 28$ pixels and allows us to use an architecture very similar to the original CapsNet [44,45].

The detailed architecture is illustrated in Fig. 9. The network has two output capsules $Z'$ (signal) and QCD (background) with two dimensions each. Inputs to the simplified model are scaled with a logarithmic function. As in Sec. 4, we train on 150,000 $Z'$ events and 150,000 QCD events with a total of 100,000 events reserved each for validation and testing. In complete analogy to the full implementation, we use a combination of margin loss for the capsules and MSE loss for the reconstruction network, where for the visualization task the reconstruction network becomes relevant. The reconstructing network achieves a classification accuracy of 95.6%, close to the approximately 96% obtained by the full network for the same problem.

In Fig. 10 we show a density plot of the two output entries in the 2-dimensional signal capsule on true signal events. Each event corresponds to one point in the 2D plane. If the classification output is proportional to the length of the capsule vector it corresponds to the distance of each point from the origin. This explains why many events are distributed in a filled circle segment distribution. A large fraction of events sits on the boundary which corresponds to the most signal-like examples. The rotation of the circle segment is not fixed a priori, each training is not guaranteed to fill the full circle, and multiple trainings will reproduce the same shape with different orientations.

In this 2-dimensional capsule plane we select five representative regions indicated by semi-transparent squares. For each region we identify the contributing events and super-impose their detector images in the $\eta - \phi$ plane in the right panels of Fig. 10. For the signal we observe bands for given rapidities and smeared out in the azimuthal angle, indicating that the network learns an event-level correlation in the two $\eta_j$ as an identifying feature of the signal. Figure 11 gives the same information for background capsule outputs on true background events. We observe the same radial pattern, but the mapping on event image reveals a very

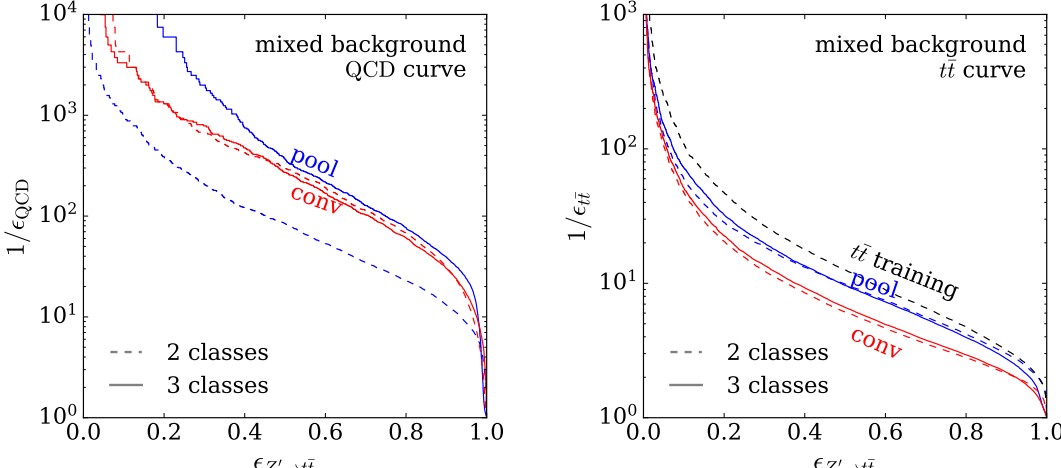

Figure 8: Left: ROC curve for the QCD di-jet background for pooling and convolutional CapsNets, each with two or three classes. Right: corresponding ROC curves for the continuum $t\bar{t}$ background. The $t\bar{t}$ training line quotes the best CapsNet result from Fig. 7. We always train on both backgrounds and only separate the testing.

different pattern with a clear back-to-back correlation in the rapidity vs azimuthal angle plane.

To better understand this behavior we transform the capsule outputs into polar coordinates, where the radius $r$ encodes the discrimination following Eq. (15) and $\varphi$ refers to different instantiations which do not matter for classification. The signal-background discriminator returns $r_S - r_B \equiv |\vec{v}^{(S)}| - |\vec{v}^{(B)}| = +1$ for maximally signal-like events and $r_S - r_B = -1$ for maximally background-like events. In Fig. 12 we first confirm that the network identifies the large jet mass for the top signal, where the secondary peak in the leading jet mass arises from cases where the jet image only includes two of the three top decay jets and learns either $m_W$ or the leading $m_{jb} \approx m_W$ [57]. Next, we see that the capsules also learn to identify the peak in the dijet invariant mass at approximately 1 TeV as identifying feature of signal events opposed to the kinematically falling spectrum for background-like events. As already observed in Figs. 10 and 11, signal jets typically have a lower separation in $\eta$ than background jets, reflected by the lower left panel of Fig. 12. Finally, we confirm that the polar angle $\varphi_S$ for signal events perfectly learns the absolute jet positions in $\eta$. We have checked that for background events the jet position in $\phi$ is learned by the corresponding background polar angle $\varphi_B$.

# 6 ttH production

Finally, we need to show how CapsNets can go beyond single images to supplement calorimeter information for example with tracking information. We illustrate this feature with one of the most complex Standard Model signatures, namely associated top-Higgs production. It allow us directly measure the Higgs-top interaction, which is, arguably, the most interesting Higgs property accessible at the LHC. The experimental challenge is that this production process comes with a low production rate and a particularly complex final state. We consider this signal combined with the dominant Higgs decay for a sizeable rate and one leptonic top decay for triggering,

$$pp \to t\bar{t}H \to t\bar{t}\,(b\bar{b})\,. \tag{18}$$

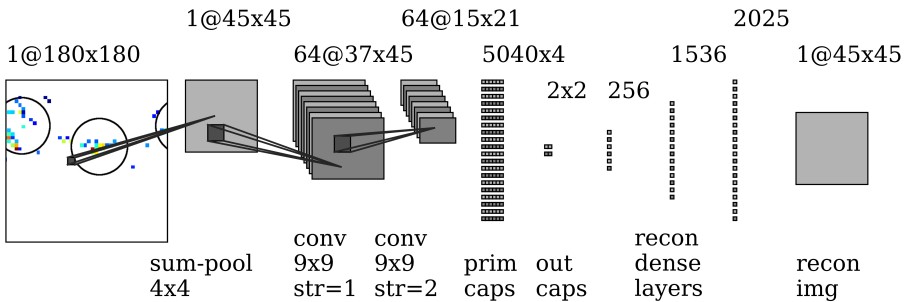

Figure 9: Simplified Capsule architecture.

The leading continuum background is

$$pp \rightarrow t\bar{t}\, b\bar{b}\,, \tag{19}$$

making the classification an ideal task for event-level machine learning and our a CapsNet tagger. An event-level Lorentz boost network has been applied to the same signal process in Ref. [74]. This network is designed to construct useful Lorentz-invariant quantities and observables from the particle 4-momenta. It is a very different approach to that considered here and serves as an excellent benchmark for our study.

We generate this process with the same setup as described in Sec. 3. We enforce decays for both processes, namely $H \rightarrow b\bar{b}$, $t \rightarrow b\ell^+\nu_\ell$ and $\bar{t} \rightarrow \bar{b}jj$ with $j = d, u, s, c$ and $\ell = e, \mu$. To analyze the event-level kinematics and relate our study to standard LHC analyses we also reconstruct jets with $R = 0.4$, even though the CapsNet analyses the calorimeter images without reference to jets. We select events with

1. exactly one muon or electron with $p_{T\ell} > 5$ GeV and $|\eta_\ell| < 2.5$;
2. at least 6 jets (anti-$k_T$ [75], $R = 0.4$) with $p_{Tj} > 20$ GeV and $|\eta_j| < 2.3$; and
3. each of the 4 $b$-jets truth-matched to a $b$-parton within $\Delta R = 0.4$.

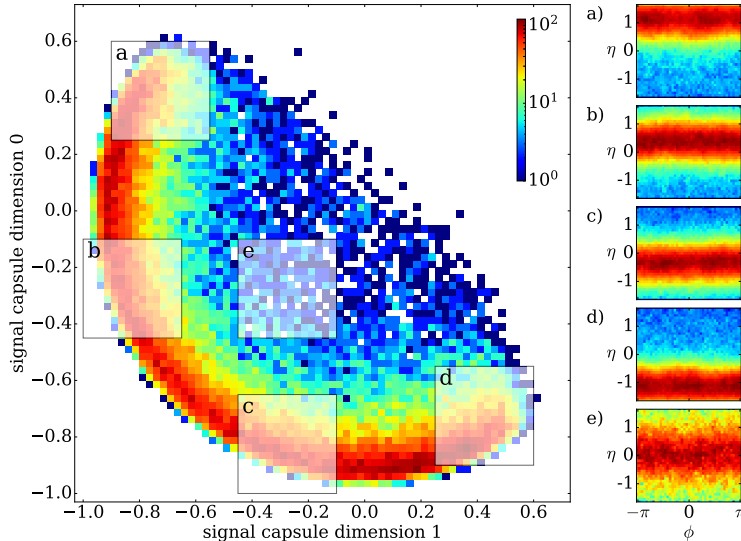

Figure 10: Distribution of the two entries in the 2-dimensional signal capsule for signal events. Right: average event images in the $\eta - \phi$ plane.

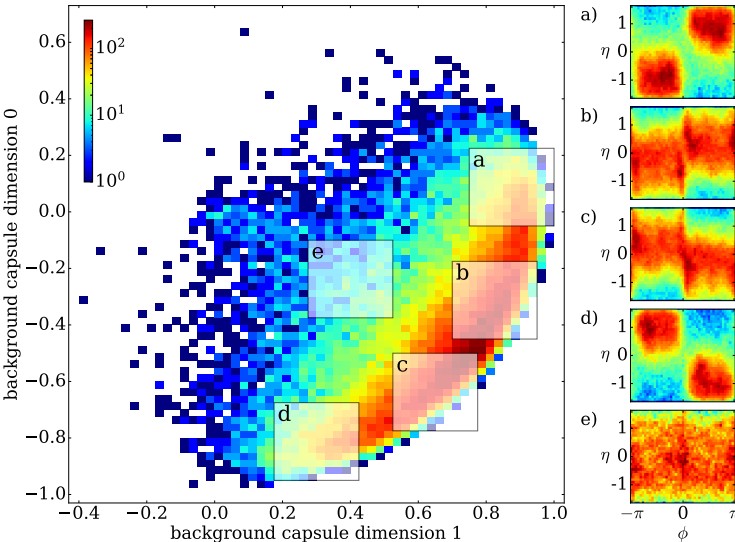

Figure 11: Distribution of the two entries in the 2-dimensional background capsule for background events. Right: average event images in the $\eta - \phi$ plane.

Because both signal and background contain four $b$-jets we do not consider a finite $b$-tagging efficiency, as it will have no significant impact on our conclusions. Assuming four $b$-tags we then reconstruct the hadronic top by combining one $b$-jet with two light-jets and minimizing $|m(j_b + j_1 + j_2) - m_t|$. Because we know that the significance is dominated by the boosted regime [56], we require the reconstructed hadronic top jet to have $p_{T j_t} > 200$ GeV and $|m_{j_t} - m_t| < 30$ GeV, to avoid producing a large number of events with little sensitivity.

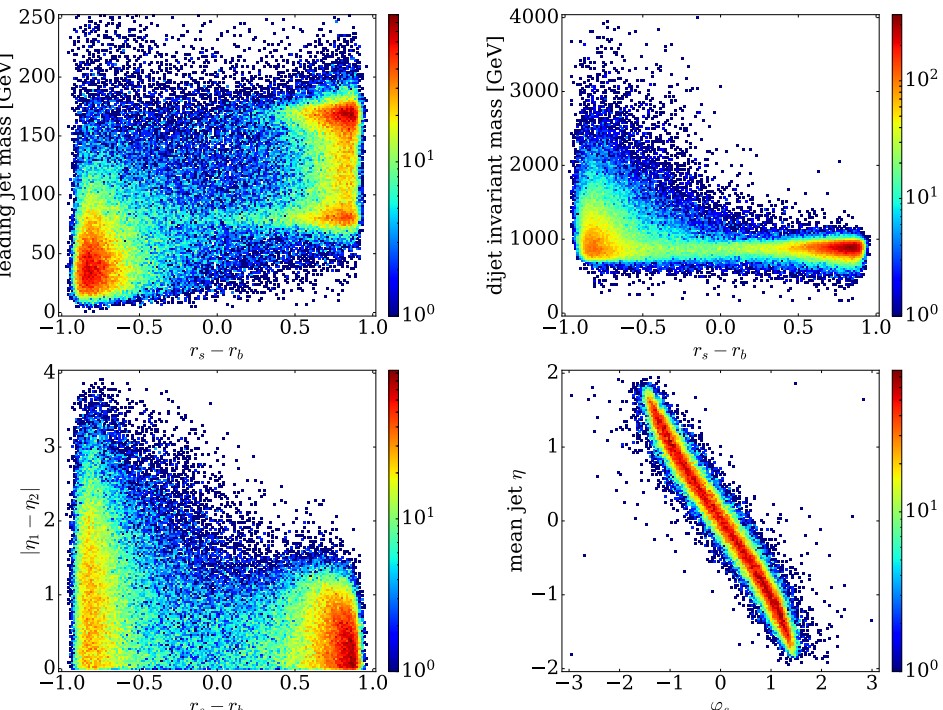

Figure 12: Correlation between capsule outputs $r_S - r_B$ and the leading jet mass, the di-jet $m_{jj}$, and $\Delta \eta_{jj}$ for true signal and background events. Finally we show the correlation of the signal $\varphi$ vs the the mean $\eta_j$ for true signal events.

Our results should not depend on this slight simplification.

For illustration, Fig. 13 shows some kinematic properties of the signal and background processes. The small differences are difficult to exploit in a cut-based analysis. A reconstruction of the Higgs mass peak is at least seriously challenging because of the $b$-combinatorics [76, 77], which is the main motivation of a boosted analysis of this process [56, 78]. To fully exploit these signal features we employ our CapsNet, to show that it can both identify objects and explore their geometric correlations.

From the previous sections we already know that we can choose a pooling or a convolutional CapsNet to analyse the event-level information for the complex $t\bar{t}H$ final state. We have seen that the convolutional CapsNet well-suited for subjet studies, but we also know that the pooling setup is superior for combining subjet and event-level information. Because the $t\bar{t}H$ analysis does not involve subjet information and the challenge will be to combine overlaying images all on the event level the convolutional CapsNet with its minimized loss of information and resolution turns out the better-suited approach.

We illustrate this CapsNet architecture in Fig. 14. We use it to analyze our usual (180×180)-pixel calorimeter image which pixel-wise encodes the $E_T$. Moreover, we want to include information from the particle identification, such as the position of identified leptons or $b$-tags. This information is included in the form of additional feature maps for each physically distinct paricle class also shown in Fig. 14 [48]. We also add a feature map with the light jet axes, which does not include any additional information but can help the network with its sparsely filled pixels. These feature maps are first combined through a 3-dimensional convolution, before each of them is independently passing through the CapsNet with its 2-dimensional convolutions. This combination of 2-dimensional and 3-dimensional convolutions allows the network to extract information both from the individual feature maps as well as correlations between them.

To understand what information the network is using for its signal vs background classification, in Fig. 15 we compare three different levels of information. First, we consider calorimeter information only, which is comparable to one of the setups in Ref. [74]. For this set-up we find comparable performance to Ref. [74], with an AUC of 0.715, which is slightly above their upper limit. The network performance is extremely poor, also because the already challenging combinatorics of $b$-jets is worsened by the many additional light-flavor jets. We can improve

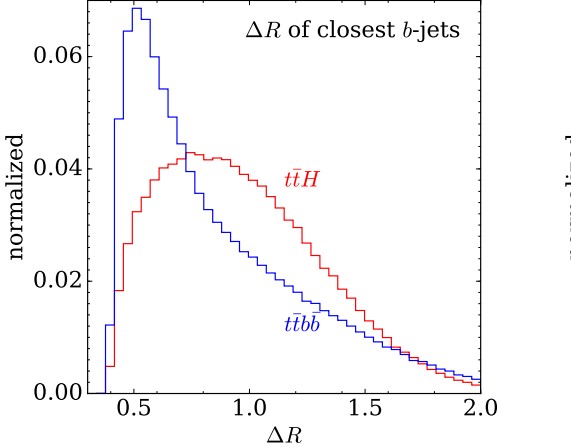
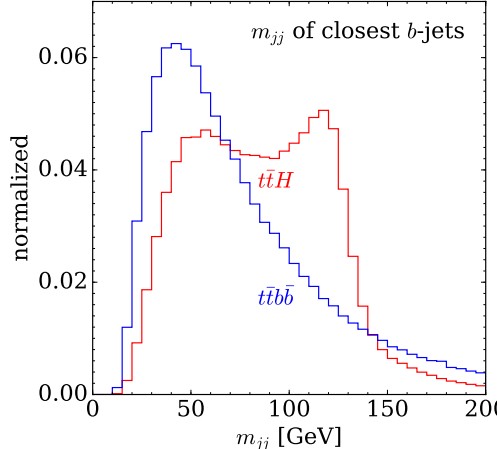

Figure 13: Left: minimum $\Delta R_{bb}$ between any two $b$-jets. Right: invariant mass $m_{bb}$ of these two closest $b$-jets.

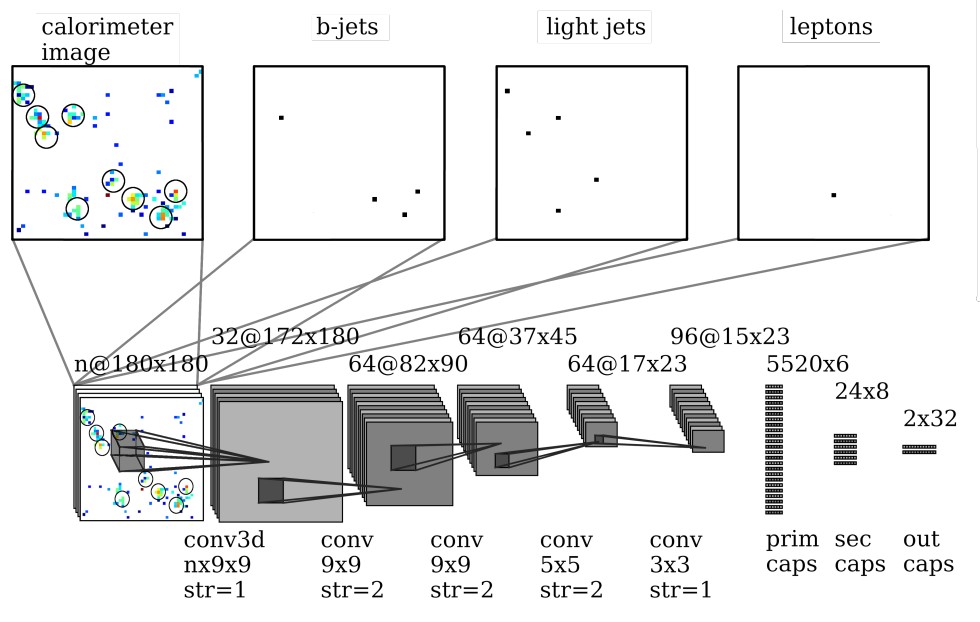

Figure 14: $t\bar{t}H$ CapsNet 3D convolution architecture with additional jet flavor and lepton information.

upon this by adding feature maps for the $b$-jets, the lepton, and potentially also the light jet axes. Figure 15 shows how this information improves the background rejection by a factor two to three and gives an area under the ROC curve of AUC=0.792. To understand where the limitations of our analysis lies and what our network is technically capable to handle we also add MC truth information. Specifically, we remove the combinatorics by labelling where each $b$-jet originates. Including this unphysical information show that our analysis is not limited by the CapsNet performance and gives us a ceiling in perormance of AUC=0.927.

## 7 Outlook

We have demonstrated the power of capsule networks for the particle physics task of LHC event tagging. Their unique representation of information makes them an ideal tool for identifying similar patterns when the convenience of regularizing images is removed.

While sparsely filled large number of pixels in calorimeter images are a limiting factor for convolutional networks, CapsNets are designed to go beyond those limitations. They are optimized to extract, both, low-level subjet information and event-level kinematics at the same time. We have illustrated the capabilities of simple CapsNets using three processes:

- tagging of a $t\bar{t}$ pair using subjet information;
- tagging and reconstructing a $Z' \to t\bar{t}$ resonance adding event-level information;
- extracting $t\bar{t}H$ production using overlaying event-level images.

We have employed different CapsNets, based on convolutions as well as based on pooling, and shown that LHC signatures typically benefit from multi-class architectures. In all of these aspects, capsules are a natural way to go beyond standard convolutional networks. Finally, we have shown how the CapsNet output is much more readily interpreted than other deep-learning architectures.

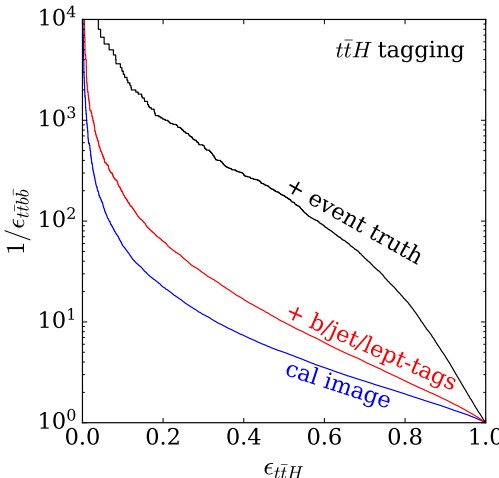

Figure 15: ROC curves for $t\bar{t}H$ with calorimeter information only, physically accessible information and with MC truth information.

# Acknowledgments

First of all, we would like to thank Anja Butter for her help during the early phase of this project. Furthermore, we acknowledge support by the state of Baden-Württemberg through bwHPC and the German Research Foundation (DFG) through grant no INST 39/963-1 FUGG. JT would like to thank BMBF for funding.

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
