# Peer review of "CapsNets Continuing the Convolutional Quest"

_SciPost Physics, doi:SciPost Phys. 8, 023 (2020)_

## Round 2 · Referee Report · Anonymous · 2019-9-9

Strengths

1 - Building up the ability of CapsNets from substructure to event-level to a combination of both.
2 - Novel visualizations in Figs 10 and 11 to see some aspect of what is being learned by the capsules.
3 - Showing how different levels of information improves $t\bar{t} H$ signal.

Weaknesses

1 - The introduction on capsules is a little confusing. It sounds like capsules must be made by a reshaping of nodes from the previous layer. However, the original paper does not construct them in this way.
2 - Not using (and not explaining why they are not used) the reconstruction part of the CapsNets. The original CapsNets paper shows that this adds to the performance.
3 - The full-event network structure clearly works, but is a complex architecture. It is unclear if it is the architecture or the capsules at the end that are allowing for the better performance.
4 - In section 6, the authors mention the Ref [43] also looks at the same signal. It is mentioned as a benchmark, but scores are not mentioned for it. While a direct comparison may not be possible, a qualitative comparison would be helpful.
5 - For openness and reproducibility, the authors could include code or code snippets for the capsule layers.

Report

Inspired by the results of S.Sabor, N. Frost, and G. E. Hinton, arXiv:1710.09829 [cs.CV], the authors show the power of CapsNets in the context of high energy physics, particularly for the LHC. They present many interesting aspects of the CapsNets, highlighting that the networks are able to take a ``full event'' and extract substructure information as along with the full event kinematics.

Overall, the paper is very strong. The authors first show that CapsNets can use jet substructure to distinguish $t\bar{t}$ samples from di-jet QCD background. Within this, they specifically remove event-level information to emphasize that the substructure is what is being used. Next, CapsNets are used to classify events as coming from either the $t\bar{t}$ continuum or through a $Z^{\prime}\rightarrow t\bar{t}$ resonance. In this case, the substructure doesn't help, but the event-level kinematics of the jets are important. From here, they add back in the QCD background It is important to note that the structure of the network for this task is significantly different than in the previous section. As a last example, they do classification for $t\bar{t} H$ vs $t\bar{t} b\bar{b}$. In the modified network for this scenario, the input images have more ``colors'' per pixel, showing that CapsNets can work like this as well.

I found this method of building up the capabilities of CapsNets to work very well. However, the specific examples that the authors chose as comparison benchmark models make it hard to see exactly how powerful the Capsules part of the network are. In the original Sabor et al. paper, CapsNets were compared with Scalar CNNs which had more standard convolutional layers than the CapsNets. In this way, they number of parameters was greatly reduced in the CapsNets and it was easy to see that the change in performance was because of the capsules. However, here that is not the case. In Fig. 4, it shows that there are 4 convolutional layers both for the CapsNets and for the reference Rutgers DeepTop Tagger. Since the convolutional aspects of the reference tagger and the new tagger are so similar, it doesn't seem surprising that the resulting ROC curves are so similar. In the remaining sections, there other reference tagger is either a simpler CapsNet, a BDT, or nonexistent. The authors could argue the power of CapsNets much better if they also included a scalar CNN (for the whole event) using a network the same as the top of Fig. 4, but replacing the primary capsules and output capsules with either more convolutions or more fully connected layers. This would exemplify the difference between the layer types.

While I just argued that the power of the capsules is not fully explored/demonstrated, I do find that Figures 10 and 11 offer a concrete example of what one can do with capsules and not with other networks. However, this was the only section that used the reconstruction part of the Sabor et al. networks. The authors never explained why this was added here. My guess is that it is the same motivation (to force the components of the capsules to contain a physical property of the event), but it is unclear. Would Figures 10 and 11 still contain the same interpretable information if the reconstruction part of the network were removed?

I also feel that one more scenario could be an interesting test case for the CapsNets. So far, the authors showed that they can do substructure, whole-event, and both at the same time. I wonder how the networks would respond to a $W^{\prime}$ with the same mass as the $Z^{\prime}$. In this case, one side would have the boosted top (identified with the substructure part of the network) but the kinematics would then be similar to the $Z^{\prime}$.

Finally, I have a few technical questions for the authors.
1) In the Sabor et al. paper, the capsules themselves are convolutional. This makes them translationally invariant. However, it seems that your capsule instead are made by just reshaping the previous layer into different capsules. Is that correct? Why was this method chosen? The squashing and routing still seems to help the capsules, but it doesn't seem as interpretable as the convolutional style.

2) The authors mention that the number of routings can be chosen and that three routing were chosen in previous studies. How many are used in these networks? Is it the same for all of the sections?

3) The Sabor et al. paper showed that the reconstruction part of the network improved the accuracy, even though it acts as a regulator. Why was it not used for most of the sections of the paper?

Requested changes

1 - Add discussion of differences between original capsule implementation and the current method.
2 - Add information about how many routings were used.
3 - More information about the preprocessing. The authors mention that CapsNets need less preprocessing, and it sounds like only scaling the images so the most intense pixel has a value of 1.0 was done. Is this the same for the benchmarks of the Rutgers DeepTop Taggers as done here?
4 - Compare CapsNets to networks of similar architecture but without the Capsules for the ``Pooling CapsNets'' architecture of Figure 6.
5 - Consider adding a $W^{\prime}$ signal to see how CapsNets deal with signal which has some substructure signals and similar kinematics. While pre-selection should be able to deal with some of the differences here, it would be for the study of the CapsNet themselves.
6 - Some mention of how the results of [43] compare to the $t\bar{t}H$ classifier used here.
7 - [Optional] Publicly available code or code snippets.

  • validity: good
  • significance: good
  • originality: good
  • clarity: high
  • formatting: excellent
  • grammar: perfect

Author:  Jennifer Thompson  on 2019-11-04  [id 639]

(in reply to Report 1 on 2019-09-09)
Category:
answer to question

Thank you for your comments and careful reading of our paper. In addition to the full response we will send formally, we would like to address point 4 in a little bit of detail here.

We compared the capsule network to a network with a similar architecture but no capsules. We include the plot here, and note that we observe a small but persistent increase in performance by using capsule networks. This comes along with the advantage of having the capsule vectors themselves, which provide a window into how the network is making decisions.

Attachment:

vsBDT_roc_curve_den.pdf

---

## Round 3 · Referee Report · Anonymous · 2019-11-14

Report

The authors have performed the checks which I asked for. I am now satisfied and happy to recommend this paper for publication.

---

## Round 3 · Author Response

We would like to thank the referee for their interest and careful reading of our paper, as well as their useful comments. We have clarified some points in our paper and addressed the points the referee has raised. Please find attached our list of changes.

---

## Round 3 · List of Changes

1 - Add discussion of differences between original capsule implementation and
the current method.
- Both implementations are, by design, identical, to clarify this we added:
``Analogous to the original capsule paper, we transition between
convolutional and capsule part by re-shaping...'' in section 4.

2 - Add information about how many routings were used.
- We used 3 routings, as was shown to be optimal in other
studies. We have rephrased a sentence to reflect this:
``We repeated this for a chosen number of routings, where
three iterations have in other studies given the best results''
now reads
``We repeated this for 3 routings, which has been
shown in other studies to give the best results''

3 - More information about the preprocessing. The authors
mention that CapsNets need less preprocessing, and it sounds like
only scaling the images so the most intense pixel has a value of 1.0
was done. Is this the same for the benchmarks of the Rutgers DeepTop
Taggers as done here?
- We have added: ``In contrast to the minimal pre-processing
we use for the event image capsule network, for the Rutgers tagger and the
jet image capsule network we employ
the full pre-processing for each jet as described in Ref. [10].
The jets are selected and centered around the $p_T$ weighted centroid of the
jet, and rotated such that the major principal axis is vertical.
The image is then flipped to ensure that the maximum activity is in
the upper-right-hand quadrant. Finally, the images are pixelated and normalized."

4 - Compare CapsNets to networks of similar architecture but without
the Capsules for the Pooling CapsNets'' architecture of Figure 6.
- We have now made this comparison, and have included the plot in a response to the report.
We observe a small but persistent increase in performance by using
capsule networks over a dense network with a similar architecture. This
comes along with the advantage of having the capsule vectors themselves,
which provide a window into how the network is making decisions.

5 - Consider adding a W′ signal to see how CapsNets deal with signal
which has some substructure signals and similar kinematics. While
pre-selection should be able to deal with some of the differences here, it
would be for the study of the CapsNet themselves.
- A W' analysis would be an interesting new application for our
network, but it would be a whole project in itself and falls
outside the scope of our current publication.

6 - Some mention of how the results of [43] compare to the t¯tH classifier used here.
- We have added ``For this set-up we find comparable performance to
Ref. [43], with an AUC of 0.715, which is slightly above their
upper limit.''

7 - [Optional] Publicly available code or code snippets.
- Unfortunately, we are unable to dedicate the time to make a public code useful to the community.

---

## Editorial Decision

published